# Scalable Model Editing via Customized Expert Networks

**Zihan Yao, Yu He,** *Tianyu Qi & Ming Li**
TAL Education Group,Beijing,China
{yaozihan1,heyu26,qitianyu3,liming9}@tal.com

## Abstract

Addressing the issues of hallucinations and outdated knowledge in large language models is critical for their reliable application. Model Editing presents a promising avenue for mitigating these challenges in a cost-effective manner. However, existing methods often suffer from unsatisfactory generalization and unintended effects on non-edited samples. To overcome these limitations, we introduce a novel approach: Scalable Model Editing via Customized Expert Networks (SCEN), which is a two-stage continuous training paradigm. Specifically, in the first stage, we train lightweight expert networks individually for each piece of knowledge that needs to be updated. Subsequently, we train a corresponding indexing neuron for each expert to control the activation state of that expert. We conducted a series of experiments on the ZsRE and Hallucination benchmarks by tuning the advanced open-source LLM, Llama2, achieving state-of-the-art results compared to current mainstream methods. Our code is available at https://github.com/TAL-auroraX/SCEN.

## 1 Introduction

With the introduction of the transformer architecture(Vaswani et al., 2017), the performance of various tasks in the field of NLP has improved rapidly, however, that has been accompanied by an explosive increase in the number of model parameters(Zhao et al., 2023; Shoeybi et al., 2019). While we are astounded by the outstanding results produced by large language models (LLMs) (Touvron et al., 2023), we also face challenges posed by their toxic and biased responses, hallucinations of factual information, and outdated knowledge (Ladhak et al., 2023; Huang et al., 2023a; De Cao et al., 2021). To address these issues, a widely adopted approach is to fine-tune LLMs for alignment, although this incurs significant computational and time costs. For LLMs deployed in industry, this issue becomes even more critical, particularly within the realm of Education, as toxic content and factual hallucinations can significantly degrade user experience(Liu et al., 2022).

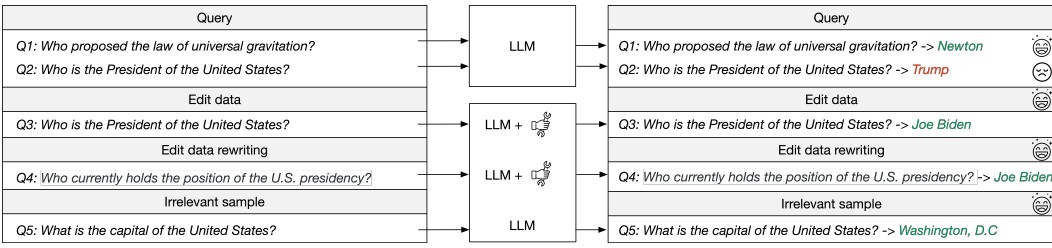

Figure 1: This is an example of model editing, where the Large Language Models (LLMs) provided an incorrect answer due to outdated knowledge (Q2). This can be corrected using model editing techniques (Q3), while ensuring a certain level of generalizability (Q4). In addition, other non-edited questions remain unaffected (Q5).

---

*Corresponding Author

To effectively address the aforementioned challenges, researchers have proposed model editing technique(Wang et al., 2023). This approach involves modifying model parameters to rectify the model's outputs, aiming to achieve desired responses for specific questions without compromising the overall accuracy for others(Fig.1). Additionally, this approach is both efficient and cost-effective. A successful editing method requires the edited model to meet three criteria(Liu et al., 2022): Reliability (rectify the outputs where the original model failed), Generality (correct response for edits rewriting), and Locality (Non-edited knowledge remains unaffected). Current mainstream methods can be roughly categorized into three types based on the location of the knowledge to be updated(Yao et al., 2023):

**Stored in an external space** (Resorting to External Knowledge): This method effectively leverages contextual information to enhance the accuracy of model response. However, it may require a substantial number of examples to achieve optimal performance. Additionally, its dependence on precise retrieval results can introduce noise.

**Stored in added parameters** (Merging Knowledge into the Model): This method directly modifies the model's parameters, enabling the integration of new knowledge with existing knowledge. However, this may lead to knowledge conflicts, and cause the forgetting of non-edited knowledge.

**Stored in the model's own parameters** (Editing Intrinsic Knowledge): Although this can achieve precise and enduring modifications to the model's knowledge, it requires a high level of understanding of the model's internal mechanisms and may lead to unforeseen side effects.

Despite some progress, these methods remain unsatisfactory due to their inherent limitations. Inspired by T-Patcher(Huang et al., 2023b), which addresses the issue of catastrophic forgetting during sequential editing by adding a series of neurons to the fully connected layer for each error-generating token, we similarly store each sample to be edited sequentially in distinct experts , and these experts are essentially components of a fully connected neural network. We then control the activation of each expert using a corresponding number of trainable neurons. Therefore, we introduce a novel two-stage continuous training model editing paradigm, SCEN (**S**calable Model Editing via **C**ustomized **E**xpert **N**etworks). Specifically, in the first stage, we train lightweight expert networks tailored to each sample that needs to be edited. In the subsequent stage, each expert is associated with an indexing neuron. These neurons are dynamically added to the network and trained using a specialized loss function during the sequential editing process. Our approach is meticulously crafted to fulfill three essential criteria for model editing:

**Reliability:** SCEN ensures reliability by assigning a customized expert to each sample that needs editing, thereby mitigates the risk of interference among different samples. The indexing neurons ensure precise activation of each expert.

**Generality:** For rewriting samples, SCEN can identify the most suitable expert by indexing neurons, which are trained using a specific margin loss function, thereby enhancing the model's generalizability.

**Locality:** The architecture of SCEN preserves all the original model weights. As long as the indexing neurons remain inactive, non-edited samples maintain their original outcomes, thereby ensuring effective localization.

In summary, SCEN presents a refined approach to model editing, achieving a balance between preserving the integrity of original knowledge and incorporating new information, while maintaining the model's robustness and adaptability. The main contributions of our work can be summarized as follows:

- We introduce SCEN, a novel two-stage continuous training paradigm for model editing. SCEN trains a customized expert network for each sample to be edited, and ensures that each expert is activated only for the current sample through a dynamic neuron indexing mechanism. SCEN is adaptable to any language model that is based on the transformer-based architecture.

- We conducted experiments on two distinct sizes of large language models, Llama2-7B/13B, focusing on two different tasks: question-answering and hallucination mitigation. The experimental results demonstrate that SCEN is an effective approach for model editing, achieving state-of-the-art performance on both tasks.

- We have conducted an in-depth exploration of the mechanisms underlying the storage of factual knowledge within LLMs, as well as the capacity of individual neurons to learn correct samples during sequence editing. This research contribution to enhancing the interpretability of LLMs.

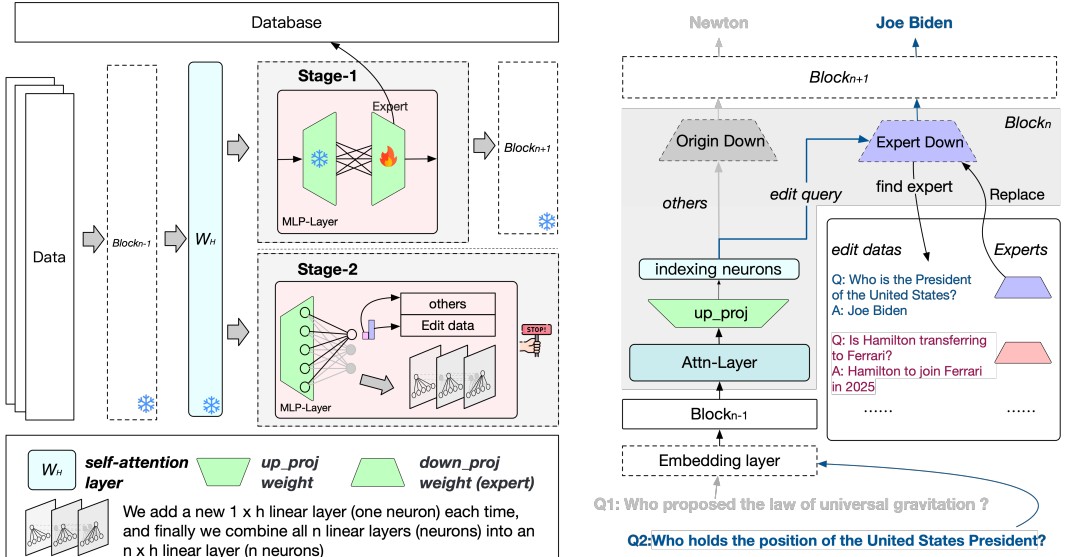

Figure 2: SCEN Overall Architecture. The left half of the figure represents the editing stages. Stage1 is the process for training the experts and Stage2 is the process corresponding to training the indexing neurons. The right half represents the inference stage, where the corresponding experts are activated by the indexing neurons to complete the subsequent inference.

## 2 Related Work

**Probing Knowledge in Pretrained Models.** Recent studies have highlighted that large language models (LLMs), such as BERT (Petroni et al., 2019) and T5 (Jiang et al., 2020; Roberts et al., 2020), effectively store and recall factual knowledge within their parameters, acting as knowledge bases. This capability is leveraged for tasks like closed-book question answering, demonstrating their potential to memorize information from extensive pre-training corpora without further fine-tuning. Probing techniques and analysis have shown that such knowledge is encoded in specific neurons, particularly within the Feed-Forward Network(FFN) layers of transformer models(Meng et al., 2022a). Researchers have identified these "knowledge neurons" at the top layers of pretrained models and have developed methods to edit factual knowledge by manipulating these characteristic FFN layers(Dai et al., 2021). This reveals a deeper understanding of the interpretability and structure of transformers, enabling more targeted and computationally efficient fine-tuning.

**Model Editing.** As mentioned in the Introduction, there are three research directions in the mainstream model editing methods, categorized by the location of new knowledge storage. Fine-tuning stands out as a straightforward and intuitive approach, which updates knowledge by updating the original model parameters. Both ROME(Meng et al., 2022a) and MEMIT(Meng et al., 2022b) exhibit enhancements derived from this perspective. This methodology preserves the original parameters and architecture of the model; however, it frequently precipitates catastrophic forgetting. To mitigate this issue, some researchers have recommended strategies that integrate external space into the model. For example, GRACE (Hartvigsen et al., 2024) adopts a novel approach by creating codebooks that store modifiable samples and employs the Euclidean distance metric to navigate the selection of specific parameter trajectories during the inference phase. Alternatively, SERAC(Mitchell et al., 2022) utilizes cache memory units to chronicle information relevant to model editing and executes

these modifications by deploying classification models for retrieval tasks. Nonetheless, these strategies necessitate the incorporation of additional storage components, which may be a limitation. Meanwhile, some researches are investigating approaches that involve adapting the model's architecture by integrating additional neurons. An example of this approach is T-Patcher(Huang et al., 2023b), which introduces new neural units for each token identified as erroneous, aiming to assimilate newly edited knowledge through an end-to-end learning process. While T-Patcher reduces the need for extra memory space and ensure the success rate of editing, it can result in an exponential increase in parameter space and computational overhead. This issue becomes particularly pronounced in the context of LLMs.

**Continual Learning(CL) in LLMs.** CL is an essential aspect of machine learning as it enables models to adapt to new tasks while retaining performance on previous ones. Recent advancements, as delineated by Lin et al. (2022), introduce an innovative paradigm known as continual model refinement (CMR). This paradigm is designed to efficiently rectify predictive inaccuracies encountered in out-of-distribution data streams, while simultaneously circumventing the peril of catastrophic forgetting. The notion of sequential model editing (SME), as expounded by T-Patcher (Huang et al., 2023b) and further developed in the work of GRACE(Hartvigsen et al., 2024), can be considered a specialized instantiation of CL. This approach entails the processing of individual data samples in a sequential manner. However, it grapples with the same challenge of catastrophic forgetting that is characteristic of broader CL approaches.

## 3 Methodology

### 3.1 Problem Formulation

Define the model $f_{model}$ as the one obtained by training on the data $D_{train}$, and we denote the subset of $D_{train}$ where $f_{model}$ makes exact predictions as $D_{loc}$. Suppose we have a model editing task using the data $D_{edit} = [d_1, d_2, ..., d_n]$, where each $d_i$ represents a sample containing input and output information $(x_i, x_i^r, y_i)$. $x_i$ is the input to be edited, and $y_i$ is the desired output. $x_i^r$ is a semantically equivalent rewrite of $x_i$, and the corresponding label for both $x_i$ and $x_i^r$ is $y_i$. We denote the sequential editing process as the time step $t \in [1, n]$, where the data required for each time step corresponds to $d_i$ in $D_{edit}$. Meanwhile, we define $f_t$ as the model obtained after editing at the time step $t$. Based on the aforementioned settings, three properties are used to evaluate the effectiveness of the sequential model editing method.

**Property 1. Reliability:** The edited model $f_t$ obtained at time step $t$, with the set of all samples up to time step $t$ denoted as $[d_1, ..., d_t]$, needs to meet the following expectations:

$$f_t(x_t) = y_t, t \in [1, .., t] \tag{1}$$

**Property 2. Locality:** For non-edited samples $D_{loc}$, $f_t$ needs to ensure that they remain unaffected:

$$f_t(x) = f_{model}(x), x \in D_{loc} \tag{2}$$

**Property 3. Generality:** $f_t$ needs to ensure that the rewritten sample $x_t^r$ also receives a correct response after editing.:

$$f_t(x_t^r) = y_t, t \in [1, .., t] \tag{3}$$

### 3.2 Customized Expert Networks

Inspired by the Mixture of Experts (MoE) architecture(Shazeer et al., 2017; Fedus et al., 2022), we propose to incorporate a series of expert networks for each incoming edit sample in a sequential editing setting. During inference, different requests will be routed to the corresponding experts to ensure optimal performance.

All transformer-based language models (LMs) are constructed using stacked transformer blocks. Assuming we are editing the $l$th layer of the LLM, and that the FFN part of $l$th layer

has two components, $W_{up}$ and $W_{down}$. We consider $W_{down}$ as a lightweight expert network, with the reason provided in the ablation studies section. We train a corresponding $W_{down}^t$ for each sample that needs to be edited. During training, we freeze all the other weights of the $f_{model}$ and activate only $W_{down}$. As shown in Stage 1 of the left side of Fig.2, after each training of $W_{down}^t$ is completed, the weights are stored in the local database, and will be used during inference when subsequent activation occurs. Since our expert networks are trained with sample-level granularity, each expert is responsible for handling a single edit sample. This ensures that there is no interference between different samples and prevents the forgetting of multiple edits.

### 3.3 Scalable Indexing Neurons

After training multiple customized expert networks, it is necessary to activate a specific expert during the inference phase to achieve the desired output. This is not a standard classification task where a sample is assigned to a specific expert. Due to the streaming nature of sequential editing, using a fixed neuron for training will inevitably lead to catastrophic forgetting. Additionally, utilizing a classification model to assign samples to experts would incur additional time costs. Inspired by T-Patcher, we propose a Scalable Indexing Neurons method, which trains a corresponding neuron for each expert using the same edit sample through a specific loss function, and these neurons are incrementally added in sequence as the editing progresses. The newly trained neurons integrated with the existing neurons and utilized during the inference phase. In the following we will provide a detailed description of how the neurons are employed during the training, merging, and inference phases.

**Training Indexing Neurons.** The settings are identical to those used for training a customized expert network with editing at $l$th layer of the LLM. We introduce a new network structure $W_{neuron}$ which is placed after $W_{up}$. This network consists of a single output neuron, implying that the parameter size of $W_{neuron}$ is $1 \times h$, where $h$ represents the dimension of FFN hidden layer. And we employ a sigmoid activation function for the output. The completed calculation is as follows:

$$\mathbf{a_t} = Sigmoid(W_{neuron}W_{up}(x_t^{att})) \tag{4}$$

where $x_t^{att}$ is the representation of the **last token** in $x_t$ obtained after the attention layer of the $l$th layer.

We determine whether a sample needs to be edited based on the activation value $\mathbf{a}$. Samples that require editing, or those that are semantically consistent with these samples, typically exhibit larger activation values, whereas non-edited samples show smaller activation values. To achieve the aforementioned objectives, we design the loss function from three aspects to control the activation value of $\mathbf{a}$ .
For Reliability and Generality, the effect is ensured by $l_{activate}$ loss, which is calculated as follows:

$$l_{activate} = exp(-\mathbf{a_t}) \tag{5}$$

where $a_t$ represents the activation value that is required at the current time step $t$ to edit the sample.

$l_{disactivate}$ loss and $l_{margin}$ loss are used to ensure Locality. We consider all the samples that need to be edited before the current time step $t$ as negative examples $a_i$, $i \in [1, ..., t-1]$. Here, we define the following two loss functions based on the activation values of these negative samples.

$$l_{disactivate} = \frac{1}{t-1} \sum_{i=1}^{t-1} exp(\mathbf{a_i} + \alpha) \tag{6}$$

$$l_{margin} = \frac{1}{t-1} \sum_{i=1}^{t-1} exp(\mathbf{a_i} - \mathbf{a_t} + \beta) \tag{7}$$

where $\alpha$ and $\beta$ are the corresponding hyperparameters, we aim to minimize the activation values corresponding to $\mathbf{a_i}$, maximize the value of $\mathbf{a_t}$, and use $\alpha$ and $\beta$ as the respective penalty terms. In our experiments, $\alpha$ was set to 0.7 and $\beta$ was set to 0.3. Therefore, the complete loss function for training indexing neurons is:

$$loss = l_{activate} + m(l_{disactivate} + l_{margin}) \tag{8}$$

$m$ is the hyperparameter as a trade-off between activation and disactivation.

Note that during the training of $W_{neuron}$, only the weights prior to the $l$-th layer of the LLM are used. Similarly, we only activate $W_{neuron}$ for gradient updates while freezing all other weights.

**Merge Neurons.** We denote the weight of the indexing neuron trained at time step $t$ as $W_{neuron}^t$. Assuming that we have edited $t$ samples, we should be able to get the weights of $t$ indexing neurons $[W_{neuron}^1, W_{neuron}^2, \dots, W_{neuron}^t]$. During inference, learn to merge these weights together to obtain $W_{merge} \in R^{t \times h}$. We define it formally as follows:

$$W_{merge} = [W_{neuron}^1, W_{neuron}^2, ..., W_{neuron}^{t-1}, W_{neuron}^t] \tag{9}$$

**Inference Stage.** Assume we have a model that has already been edited at $l$th layer. Now, given a new test sample $q$ , how can we obtain the activation values of the indexing neurons for this sample? When the forward inference reaches the $l$th layer, we can obtain the answer using the following formula. It should be noted that here we take the last token of the input $q$ as the representation of the sample.

$$\mathbf{a^q} = Sigmoid(W_{merge} W_{up}(x_q^{att})) \tag{10}$$

where there are $t$ activation values in the $\mathbf{a^q}$ vector, each corresponding to an experts, denote $a_i^q$ as the activation value at the $i$th position in $\mathbf{a^q}$. Here, we determine whether to activate an expert or use the model's original weights by setting a threshold $\theta$.
Following the rule described below, as shown in Eq.(11). When the activation value is below the threshold $\theta$, the model's original weights $W_{down}$ are used, without activating any experts. Conversely, if the maximum activation value $a^q$ exceeds the threshold, the index corresponding to this maximum value will indx to a unique expert, which will be used to replace the original weights, as shown in Eq.(12). After determining the structure of the $FFN^l$ at the $l$th layer, the inference proceeds forward. The complete inference process is illustrated in the right half of Fig.2.

$$FFN^l = \begin{cases} W_{down}^{Expert(a^q)} W_{up} & \text{if } \max(a^q) > \theta, \\ W_{down} W_{up} & \text{otherwise,} \end{cases} \tag{11}$$

$$Expert(a^q) = \arg\max_i a_i^q \tag{12}$$

## 4 Experiments

### 4.1 Experimental Setup

**Datasets.** We utilized two benchmark datasets suitable for sequence editing, ZsRE and Hallucination, to evaluate the effectiveness of our method. ZsRE is a comprehensive question-answering (QA) dataset(Levy et al., 2017). We randomly selected 20,000 entries from this dataset. Specifically, 10,000 entries,denoted as $D_{train}$, were used for typical supervised fine-tuning (SFT) based on the LLaMA2 model, while the remaining 10,000 entries, denoted as $D_{test}$, were reserved for evaluating the performance of the SFT model. Based on the evaluation results, we divided the post-test data into two subsets: $D_{edit}$ and $D_{loc}$. The

subset $D_{edit}$ consists of samples where the SFT model made incorrect predictions. From this subset, we selected the first 200 and first 1000 samples for subsequent editing operations. Conversely, $D_{loc}$ comprises samples where the SFT model made correct predictions. Similarly, we selected the first 1000 samples from this subset to evaluate our method's Localization metric. Detailed information is provided in Appendix A. **Hallucination** is a dataset designed to evaluate the performance of model editing methods in of mitigating model hallucinations, as described by Manakul et al. (2023) and (Hartvigsen et al., 2024). Manakul et al. (2023) prompt GPT-3 to generate 238 wikipediastyle biographies for concepts extracted from WikiBio, they then annotate factual accuracy of each sentence, noting which are hallucinations. We followed the processing method described by (Hartvigsen et al., 2024) and created 1392 sequential edits and 592 accurate outputs. Based on the WikiBio dataset, we incorporated 200 samples from OpenWebText and trained Llama 7B and 13B respectively. During the editing phase, we selected the first 200 out of the previous 1392 sequential edits for evaluation, denoted as WikiBio-E. Additionally, we used the results from 516 accurate outputs, referred to as WikiBio-A, combined with 200 samples from OpenWebText as a test set to evaluate the localization.

**Baselines.** We compare the proposed method with several mainstream model editing methods. Fine-tuning based methods (including FT and FT+KL), **FT** refers to the scenario where the fully connected network within a certain transformer block is trainable,while, **FT+KL** incorporates the Kullback-Leibler (KL) divergence into the loss function, as described in (Mitchell et al., 2021), to mitigate catastrophic forgetting. Memory-based methods, such as **SERAC**(Mitchell et al., 2022) and **GRACE** (Hartvigsen et al., 2024), have shown significant promise. GRACE, in particular, is the latest state-of-the-art (SOTA) method based on memory. **MEND** (Mitchell et al., 2021) aims to learn a HyperNetwork using additional training data to transform the gradients obtained through standard fine-tuning. **MEMIT**, as proposed by (Meng et al., 2022b), is a method for directly updating a language model with many memories.In the realm of generative tasks for LLMs, T-Patcher(Huang et al., 2023b) introduces neurons at the token level. Inspired by this approach, we explore the addition of neurons at the granularity of samples. However, due to the inherent characteristics of **T-Patcher**, it exhibits substantial training and inference performance constraints. Consequently, we did not experiment with T-Patcher on the Hallucination dataset nor perform a thousand edits on the ZsRE dataset. Detailed parameter settings and experimental details for each method are reported in Appendix C.

**Metrics.** We used Accuracy and Perplexity to evaluate the ZsRE and Hallucination datasets. In ZsRE, the three metrics **Reliability**, **Generality**, and **Locality** refer respectively to whether all edited samples are successful, whether rewrited samples can answer correctly, and whether non-edited samples can answer correctly. In Hallucination, **WikiBio-E**, **WikiBio-A**, and **OpenWebText** refer to whether the perplexity of the edited samples is reduced and whether the perplexity of the unrelated WikiBio-A and OpenWebText can be maintained at a lower level, respectively.

## 4.2 Experimental Results

We conducted 200 and 1000 sequential editing experiments on Llama-2 7B and 13B respectively, as shown in Tables 1 and 2, respectively. For the ZsRE dataset. We focus on two aspects: the trade-off between Reliability and Locality, and the average performance across three metrics. For the Hallucination dataset, we examine whether the ppl of WikiBio-E decreases, and whether the ppl of WikiBio-A and OpenWebText increases due to halluciation effects. The **FT+KL** method could not be evaluated on the Hallucination dataset due to it's requirement for additional data.The **MEMIT** method is not a sequential editing method and performs significantly below baseline levels in hallucination scenarios, so it is not used in our comparison.

On the ZsRE dataset, both at 200 and 1000 edits for long sequences, Tables 1 and 2 demonstrate that our proposed method(**SCEN**) substantially outperforms the comparison methods on average across all three metrics. In terms of Generality and Locality, SCEN achieves a better trade-off on both the Llama2 7B and 13B models, and it also demonstrates improved stability. In contrast, the direct fine-tuning(**FT**) method demonstrated that while

| Method | ZsRE(ACC ↑) | | | | Hallucination(PPL ↓) | | | |
|---|---|---|---|---|---|---|---|---|
| | Reliability | Generality | Locality | *Avg.* | WikiBio-E | WikiBio-A | OpenWebText | *Avg.* |
| *Llama2-7B* | | | | | | | | |
| FT | 98.0 | **93.0** | 48.6 | 79.9 | **1.25** | 2.58 | 5.31 | **3.04** |
| FT+KL | 92.0 | 83.8 | 55.7 | 77.2 | - | - | - | - |
| MEND | 1.0 | 2.83 | 96.7 | 33.5 | 19.3 | 2.33 | 4.82 | 8.82 |
| SERAC | 89.0 | 16.2 | 81.8 | 62.3 | 20.6 | 2.31 | **4.79** | 9.23 |
| MEMIT | 24.0 | 39.9 | 17.0 | 27.0 | - | - | - | - |
| T-Patcher | 94.0 | 87.9 | 62.9. | 81.3 | - | - | - | - |
| GRACE | 94.5 | 38.2 | **99.9** | 77.5 | 3.67 | 2.31 | **4.79** | 3.59 |
| SCEN | **100.0** | 90.0 | 83.3 | **91.1** | 2.93 | **2.28** | 4.82 | 3.34 |
| *Llama2-13B* | | | | | | | | |
| FT | 93.5 | 86.7 | 48.8 | 76.3 | **1.10** | 2.40 | 5.94 | 3.15 |
| T-Patcher | 92.5 | 85.3 | 55.8. | 77.9 | - | - | - | - |
| GRACE | 90.0 | 40.6 | **100** | 76.9 | 2.21 | 1.98 | **4.69** | 2.96 |
| SCEN | **99.5** | 80.7 | 83.5 | **87.9** | 1.57 | **1.97** | 4.70 | **2.75** |

Table 1: Comparison of SCEN with existing methods. All experimental results were obtained after 200 edits.

| Method | Llama2-7B (ACC ↑) | | | | Llama2-13B (ACC ↑) | | | |
|---|---|---|---|---|---|---|---|---|
| | Reliability | Generality | Locality | *Avg.* | Reliability | Generality | Locality | *Avg.* |
| FT | 92.1 | **87.3** | 38.7 | 72.7 | 90.8 | **83.0** | 34.9 | 69.6 |
| GRACE | 90.0 | 33.8 | **99.9** | 74.5 | 92.6 | 38.2 | **100.0** | 76.9 |
| SCEN | **96.2** | 80.2 | 83.8 | **86.7** | **98.0** | 76.1 | 70.2 | **81.4** |

Table 2: Results were obtained by performing 1000 sequential edits on the ZsRE dataset.

Reliability and Generality were maintained, there was a significant drop in Locality. To address this limitation, **FT+KL** was utilized; however, its impact on improving Locality was minimal. The **GRACE** achieved the best results on Locality but its Generalization was very insufficient. Similarly, **SEARAC** exhibited an imbalance between Generality and Locality. **T-Patcher** achieves results close to SCEN on the Reliability and Generality metrics. However due to the addition of neurons for each token which significantly increases the number of parameters and the fact that **T-Patcher** is an end-to-end approach, these factors collectively contribute to its unsatisfactory performance on the Locality metric. **MEMIT** is not a sequential editing approach, and thus it does not achieve superior results across all three metrics. Overall, we propose the dynamic addition of indexing neurons, which can accurately activate experts for long sequence edits without catastrophic forgetting. This method strikes a balance between Generality and Locality without resorting to extremes.

On the Hallucination dataset, it can be seen from Table 1 that **FT** is an effective method that drastically reduces the perplexity of editing samples. Due to its excellent performance on WikiBio-E, FT achieved the best result on the Llama2-7B model, with SCEN ranking second. On the other hand, SCEN achieved the best results on the Llama2-13B model, maintaining a low level of average perplexity across the three test sets. Meanwhile, GRACE was competitive and secured second place. In summary, it can be seen that SCEN also maintained a relatively good stability in the hallucination relief task.

### 4.3 Ablations and Analysis

**Ablation Study.** We conducted two ablation experiments in SCEN to investigate the contribution of each component. Specifically, we examined the impact of using different positions of weights as experts for editing and employing different training loss functions for indexing neurons. The complete experimental results are shown in Table.3, where the top three rows illustrate the effect of editing on different positions of weights. $W_{attn}$ indicates that editing is performed in the attention layer, $W_{up}$ indicates that editing is performed in the upper half of the fully-connected layer following the attention layer, and the corresponding $W_{down}$ indicates that editing is performed in the lower half of the fully-connected layer. For a more concise description of $W_{attn}$, $W_{up}$ and $W_{down}$ , please refer to

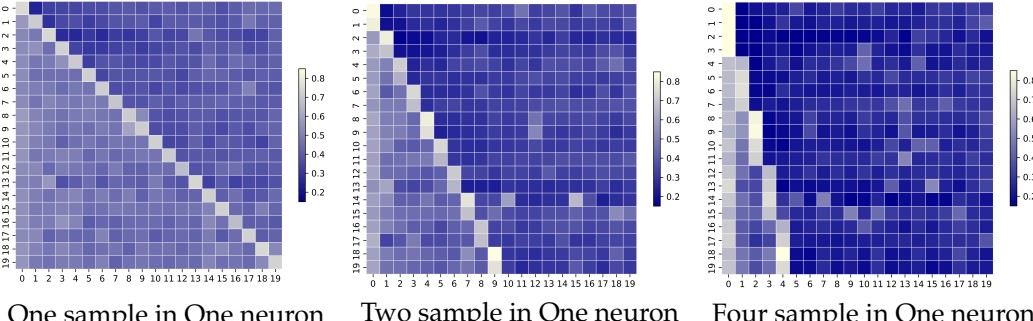

One sample in One neuron    Two sample in One neuron    Four sample in One neuron

Figure 3: The Impact of Varying Sample Sizes on the Activation Patterns of Indexing Neurons.

Fig.2. The following three rows in Table.3, please refer to the effect of employing different training loss functions for indexing neurons on the final results. w/o $l_{disactivate}$&$l_{margin}$ represents the scenario in which we retain only the activation loss, while w/o $l_{margin}$ represents retaining both the activation loss and the disactivate loss. It can be observed that in terms of the positioning of expert weights, $W_{down}$ achieves the optimal performance in editing, whereas $W_{attn}$ exhibits the poorest performance. These findings are consistent with the results reported by ROME(Meng et al., 2022a). In terms of loss, the main contribution is attributed to $l_{disactivate}$, while the effect of addition of $l_{margin}$ also shows improvement.

| | Llama2-7B (ZsRE) | | | |
|---|---|---|---|---|
| | Reliability | Generality | Locality | *Avg.* |
| $W_{attn}$ | 90.0 | 77.3 | 92.0 | 86.4 |
| $W_{up}$ | 96.5 | 83.2 | 86.7 | 88.8 |
| $W_{down}$ | 100.0 | 90.0 | 83.3 | 91.1 |
| w/o $l_{disactivate}$&$l_{margin}$ | 100 | 93.8 | 73.7 | 89.2 |
| w/o $l_{margin}$ | 100 | 90.6 | 82.1 | 90.9 |
| SCEN loss(Eq.(8)) | 100 | 90.0 | 83.3 | 91.1 |

Table 3: Ablation Experiments of SCEN on the ZsRE Dataset

**How many samples can each expert handle?** We consider whether the added parameters can be reduced in terms of optimizing storage space and inference time. We conducted three sets of experiments in which an expert and its corresponding indexing neuron were responsible for handling one sample, two samples, and four samples, respectively. In this way the present, after sequential editing 200 entries, we obtained 200, 100, and 50 experts, respectively. With Table.4 shows that the number of added parameters has decreased, and both Reliability and Generalizability metrics have significantly declined. This decline may be attributed to each expert learning from multiple diverse training samples, which consequently leads to a significant reduction in the robustness of indexing neurons.

It is evident that a single neuron is insufficient to accurately model multiple diverse samples. We also demonstrate the activation of the corresponding indexing neurons for different numbers of samples, as illustrated in Fig.3. The horizontal coordinates represent the indexing neurons corresponding to each edited expert, while the vertical coordinates represent the sequentially edited samples. In the left figure, as it is a one-to-one sample, clear highlighting can be observed along the diagonal. The middle and right figures illustrate the two-to-one and

| | Llama2-7B (ZsRE) | | | |
|---|---|---|---|---|
| Numbers | Reliability | Generality | Locality | Experts |
| One | 100 | 90.0 | 83.3 | 200 |
| Two | 86.5 | 72.7 | 81.7 | 100 |
| Four | 53.0 | 43.7 | 84.7 | 50 |

Table 4: Performance of Different Number of Compression Strategies on ZsRE

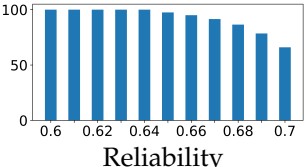 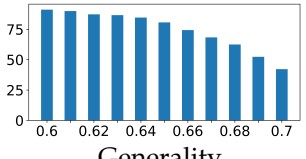 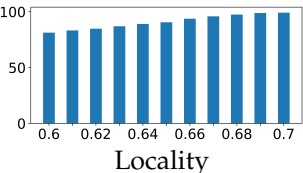

Figure 4: Impact of $\theta$ Value Variations on the Performance of the ZsRE Dataset

four-to-one scenarios, respectively, showing that the activation state progresses in a stepwise manner.

**Activation parameter analysis.** We investigated the effect of the size of the activation threshold value $\theta$ on the final Reliability, Generality and Locality by dividing $\theta$ into 11 hyperparameters ranging from 0.60 to 0.70 in increments of 0.01.The experiments were conducted on the ZsRE dataset, as illustrated in Fig.4. It can be observed that lower activation values favor Reliability and Generality, but Locality suffers more. The trend indicates that Reliability and Generality decrease with increasing $\theta$, while Locality increases with increasing $\theta$ . Memory-based model editing approaches are frequently challenged by the need to balance Generality and Locality. SCEN minimizes the situations where hyperparameters significantly impact the metrics. As shown in Fig.4, our method exhibits relatively small fluctuations across all three metrics within the first half of the thresholds.

**Which layer is better for editing?** We examined the performance of the even-numbered layers of Llama2-7B separately, as shown in Fig.5. It is evident that in the lower-level transformer blocks of Llama2-7B, editing has minimal effect, thereby preserving the original model results and maintaining a high level of Locality. In contrast, in the higher-level transformer blocks, the impact of editing is pronounced. Starting from the 16th layer onward, Reliability remains at a high level, with minimal fluctuations in Generality and Locality. Consequently, we further validate that the higher-level transformer blocks of LM , based on the transformer architecture, contain some factual knowledge, and editing of these layers has a significant effect.

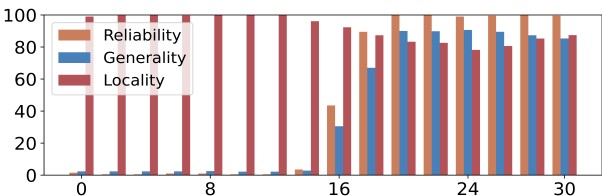

Figure 5: Results of Editing with SCEN at Even-Numbered Layers of Llama 2-7B

## 5   Conclusions

In this paper, we present SCEN, an innovative two-stage continuous training paradigm for model editing. This approach involves training a customized lightweight expert network for each sample to be edited. Additionally, we have developed a scalable dynamic neuron indexing mechanism that efficiently activates the corresponding experts. Our experimental results indicate that SCEN consistently surpasses the most latest state-of-the-art methods in model editing. Furthermore, we delve into the interpretability of LLMs with a focus on transformer-based architectures, substantiating that factual knowledge is predominantly stored in the latter layers of the language model. Looking ahead, we plan to explore advanced weight compression techniques aimed at reducing the number of expert networks and indexing neurons. This effort should lead to faster processing and less storage use, making model editing more practical and efficient.

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

## A    ZsRE dataset details

For ZsRE, we first trained the base model using 10k data to get an SFT model. The format of the data is shown in Table 5. We added [INST] and [/INST] to the query, which is to simulate the chat model training format of LLM. The format in editing and result evaluation is consistent with the format of the training data.We additionally sampled 10k samples $D_{test}$ for constructing edit samples $D_{edit}$ and samples $D_{loc}$ for evaluating localization. $D_{edit}$ is the sample where the SFT model answered incorrectly in $D_{test}$.In the experiments of this thesis, we took the first 200 and the first 1,000 samples of $D_{edit}$, respectively, which were used to evaluate the effect of editing, meanwhile, we used the first 3 rewrites of these samples as a test set to evaluate the generalizability.

| Prompts | Answer |
|---|---|
| [INST]Whose direction is From Dusk till Dawn?[/INST] | Robert Rodriguez |
| [INST]The date of birth for Sinoti Sinoti is what?[/INST] | 9 September 1985 |
| [INST]In which war did Paul Lacombe de La Tour fight?[/INST] | World War I |
| ...... | |

Table 5: Data formats for training SFT models in ZsRE

## B  Hallucination dataset details

GRACE constructed a dataset via SelfCheckGPT(Manakul et al., 2023) to evaluate the effect of model editing methods to reduce model illusions, and called this dataset **Hallucination**. Manakul et al. (2023) utilizes GPT3 to extract concepts from WikiBio to generate corresponding biographies. For each sentence generated, they labeled which sentences were factual and which were hallucinatory. Follow the idea of GRACE to edit the highly inaccurate sentences and replace them with real Wikipedia sentences. The highly accurate samples are utilized to judge whether the model still maintains a relatively low level of perplexity on these samples after editing. Through the above processing, a total of 1,392 editing samples and 516 sentences that have been accurate have been generated. Compared with the dataset used in the previous model editing, Hallucination has a longer text length, which is more in line with the data situation in the real scenario. In this paper, we used the first 200 of the 1392 edited samples for our experiments, in the two Llama2-7B/13B models. Partial data are shown in Table 6.

---

**WikiBio-W:**
This is a Wikipedia passage about akila dananjaya. Akila Dananjaya (born 2 August 1995) is a ......
This is a Wikipedia passage about wilhelm windelband. Wilhelm Windelband (15 March 1848 ......

......
**WikiBio-A:**
This is a Wikipedia passage about rick mahler. Rick Mahler (born Richard Alan Mahler on April .....

......
**OpenWebText:**
A magazine supplement with an image of Adolf Hitler and the title 'The Unreadable Book' is pictured .......
For today's post, I'd like to take a look at California's voter initiative to legalize pot. If the measure ......
......

---

Table 6: Data format for continued training in Hallucination

## C  Experiment details

Detailed parameter settings using the comparison method used are described in detail below:

For **FT+KL**, additional data is needed to be used with the *KL* loss. In the ZsRE data, we should have a vector indexing model(all-mpnet-base-v2) for each edit sample from the 5 most similar samples found in the training data of SFT. The training step for each edit sample is 10 and the learning rate is 0.0001.

For the **MEND**, additional data is needed to train an editor, and to avoid any test set leakage, in the ZsRE data, we use 2000 entries from the data used to train the SFT to train the editor. In the Hallucination dataset, we use the last 400 data out of 1352 entries to train the editor.

For the **SEARAC** method, the method contains a total of three models, of which the classification model we use is distilbert-base-cased, and the threshold we set is 0.6.

For the **GRACE** method, $\epsilon$ is set to 3.0, the number of iterations is set to 10, the learning rate is set to 1.0, and the adapter weights are taken to be randomly initialized.

For **MEMIT**, we only experimented on ZsRE, the fact token used makes the last token of the entity, and the positions to be edited are the 20th and 22nd transofmerblock.

