# OpenReview forum: "Scalable Model Editing via Customized Expert Networks"
_colmweb.org/COLM/2024/Conference — COLM_

### Official Review · Reviewer_4Ci8 · 2024-05-09

**Rating:** 6
**Confidence:** 4
**Ethics Flag:** 1

**Summary:**

This paper presents an LLM editing method using customized expert networks and indexing neurons.  Experiments have been conducted to demonstrate the effecitveness of the proposed method.

**Reasons To Accept:**

The idea of using  customized expert networks for model editing seems novel to me.

**Reasons To Reject:**

The evaluation part of this paper can be further improved.
1) Some ablation studies are expected to analyze the effectiveness of different component of the proposed method, such as the different losses in Eq.(8).
2) How about the space and time efficiency of the proposed method since a separate expert network is necessary for each edited knowledge?

---

> ### Author Rebuttal · Authors · 2024-05-28
>
> Thanks for your valuable reviews, we hope the following comments could address your concerns
> > 1. Some ablation studies are expected to analyze the effectiveness of different component of the proposed method, such as the different losses in Eq.(8).
>
> Indeed, conducting ablation studies is a crucial step in understanding the impact of each individual component within our proposed method. In light of the first question you posed, we have carried out additional experiments using the llama7B model on the ZsRE datasets for 200 edits. The results of these experiments are presented in the table below:
> | | | ZsRE Llama2-7B| | | |
> |--|--|:--:|--|--|--|
> | | | Reliability|Generality|Locality|Avg|
> | 1 | w/o $l_{disactivate}$ & $l_{margin}$ | 100 | 93.8 | 73.7|89.2|
> | 2 | w/o $l_{margin}$ | 100 | 90.6 | 82.1| 90.9|
> | 3 | SCEN loss (Eq. 8)  | 100 | 90.0 | 83.3 | 91.1 |
>
> We have drawn the following conclusions: By comparing w/o $l_{margin}$ and SCEN, we observe that the main contribution of our proposed loss comes from $l_{disactivate}$ , while adding $l_{margin}$ still improves the method’s performance.
> > 2. How about the space and time efficiency of the proposed method since a separate expert network is necessary for each edited knowledge?
>
> We train a customized expert network for each edited knowledge and store the weights in memory. During the inference process, only a single active expert weights is loaded into the GPU's VRAM , ensuring that the VRAM utilization remains controlled. The primary trade-off involves the time required to transfer the weights from memory to VRAM, which takes about 18ms.
>
> Certainly, to provide a more intuitive comparison between  SCEN and T-patcher, which is also designed for editing sequence models, we performed a analysis focused on the differences in VRAM consumption and time overhead. This analysis was based on 1000 editing operations on Llama-7B using the A800（80G）GPU, as illustrated in the table below：
> |  | Ex-GPU(M) | In Edits(s) | Out Edits(s) |
> |--|--|--|--|
> | T-patcher | 121.664 | 127.7 | 126.76 |
> | SCEN  | 20.996| 303.5  | 125.32 |
>
> In summary, T-patcher is less time-consuming for edited datasets but comes with a higher GPU memory cost. SCEN is more memory-efficient but takes longer to process edited datasets. Both methods have comparable performance for unedited datasets. The conclusion is consistent with the analysis above.
>
> All of the above modifications will be incorporated into the camera-ready version.

---

> > ### Author Response · Authors · 2024-06-06
> > **Request for response**
> >
> > Dear reviewer 4Ci8,
> >
> > We have posted extra experiment results (e.g., a supplemented baseline and other ablation studies) and clarifications to answer your questions. We wonder if you can let us know whether our responses address your concerns.
> >
> > Look forward to your reply.
> >
> > Best regards,
> >
> > Authors

---

### Official Review · Reviewer_5GUJ · 2024-05-09

**Rating:** 5
**Confidence:** 4
**Ethics Flag:** 1

**Summary:**

The paper proposes a method for continual model editing called SCEN. Like GRACE, it only modifies the l-th layer (a hyperparameter), specifically the down projection matrix, for each edit t which is then stored in a database (an "expert"); after T edits, at test time, the model must decide which expert to use (or none). GRACE does this by codebook maintenance, which requires no further training. SCEN proposes to do this by training an expert selector. There are presumably many ways to parameterize the selector, but the paper loosely follows the training scheme in T-Patcher to train a scalar projection for each edit to activate for that edit (by contrastive loss), then at test time uses the argmax of the projection for the given query if the value is above some threshold.

Experiments are conducted on ZsRE (QA) and GRACE's version of hallucination task (reducing perplexity on edits while maintaining it on non-edits), using Llama-2. SCEN performs better than GRACE on edit success and generalization, but worse on locality; on average SCEN performs better.

**Reasons To Accept:**

- A reasonable modification of GRACE (vector codebook -> down projections, Euclidean distance -> trained expert selector). The method is also simple.
- Better performing than GRACE in the average over reliability/generalization/locality.

**Reasons To Reject:**

- Somewhat limited novelty, which is a general problem with many variations of model editing methods (slight differences in parameterization/activation approach, which are often almost certainly over-optimized for small-scale models and datasets).
- No comparison with T-Patcher, which doesn't make sense to me. It has the same continual model editing setup, and this paper is inspired by it.
- More about T-Patcher: it bakes in the activation for the right expert into training. So even though it doesn't explicitly predict the expert at inference time (i.e., the neurons are not stored in a database but part of the model), it's kind of doing the same thing? T-Patcher is supposed to be also scalable (i.e., the added neurons are a minor portion of the model for thousands of edits).
- Limited tasks: why only ZsRE and Hallucination, when the main baseline GRACE also includes SCOTUS?
- Some writing issues (unnatural)

---

> ### Author Rebuttal · Authors · 2024-05-29
>
> >1.Somewhat limited novelty...
>
> I'd like to highlight the distinct contributions of our two-stage editing approach.
>
> SCEN offers a targeted and efficient way to correct a finite set of erroneous knowledge, which is crucial for preserving the accuracy and relevance of models in a real online environment. It ensures that, as the number of edits increases, the original model's capabilities do not degrade.
>
> >2.No comparison with T-Patcher...
>
> Indeed, it was an oversight on our part not to include a direct comparison with T-Patcher initially. To rectify this, we have now implemented T-Patcher's method on both the llama2 7b and 13B models on the ZsRE datasets for 200 new edits. The table below presents the results:
>
> | |Reliability|Generality|Locality|Avg|
> |--|--|--|--|--|
> |T-Patcher-7B|94.0|87.0|62.9|81.3|
> |SCEN-7B|100|90.0|83.3|91.1|
> |T-Patcher-13B|92.5|85.3|55.8|77.9|
> |SCEN-13B|99.5|80.7|83.5|87.9|
>
> The results indicate that our method, SCEN, maintains a good balance between Reliability, Generality, and especially Locality, compared to T-patcher.
>
> >3.More about T-Patcher...
>
> Our SCEN method is like T-Patcher because both keep editing knowledge in network weights, but they work in different ways. In SCEN, the experts are stored in memory and only the activated expert is loaded onto the GPU, which optimizes memory usage and ensures precise expert activation. In contrast, T-Patcher's neurons, which are integrated into the model, are always active during inference, leading to increased GPU consumption and potential inaccuracies in neuron activation. You can find the experiment for GPU usage from 4Ci8. And for cases where neurons are activated incorrectly, see the table below:
> | | False Activation(%) |Correct Response(%)|
> |--|--|--|
> | T-patcher | 71 | 48.5|
> | SCEN | 7.5 | 93.5 |
>
> The results indicate that SCEN has a higher acc rate and a lower rate of false activations for non-edit data.
>
> >4.Limited tasks...
>
> Historically, models like GRACE were tested on Encoder-based frameworks such as BERT, which are suited for classification tasks like SCOTUS for performance evaluation. However, with the emergence of LLM, the research focus has shifted towards these Decoder-only models. Our approach is primarily based on LLaMA, and therefore does't incorporate traditional classification tasks.
>
> >5.Some writing issues.
>
> We apologize for any trouble caused by the writing issus and are dedicated to carefully revising and correcting them to ensure a more natural final version.

---

> > ### Author Response · Authors · 2024-06-06
> > **Request for response**
> >
> > Dear reviewer 5GUJ,
> >
> > We have posted extra experiment results (e.g., a supplemented baseline for T-Patcher) and clarifications to answer your questions. We wonder if you can let us know whether our responses address your concerns.
> >
> > Look forward to your reply.
> >
> > Best regards,
> >
> > Authors

---

### Official Review · Reviewer_amby · 2024-05-11

**Rating:** 6
**Confidence:** 3
**Ethics Flag:** 1

**Summary:**

The paper presents SCEN, a two-stage model editing approach for large language models that improves upon outdated knowledge and hallucinations. SCEN stands out for its custom expert networks per knowledge piece and indexing neurons that control expert activation, leading to state-of-the-art results in question-answering and hallucination reduction datasets.

**Questions To Authors:**

Please refer to the section titled 'Reasons for Reject' for further details.

**Reasons To Accept:**

* The paper introduces a novel method for model editing that significantly improves upon existing techniques.

* The two-stage continuous training paradigm is promising, combining the benefits of specialized expert networks with a dynamic neuron indexing mechanism to apply edits selectively.

**Reasons To Reject:**

* While you have mentioned T-Patcher, it is essential to provide a comprehensive overview of these related works and highlight the differences between your paper and theirs.

* Consider including an experimental discussion on the effects of fine-tuning the attention layer or the $W_{up}$​ weight. This exploration will provide valuable insights into the performance improvements achieved by these modifications.

* In Tables 1 and 2, the model performance based on Llama2-13B as the baseline is generally much worse than that based on Llama2-7B. The authors could explain or reasonably speculate on the reasons behind this.

---

> ### Author Rebuttal · Authors · 2024-05-29
>
> > 1.While you have mentioned T-Patcher...
>
> SCEN and T-patcher are both innovative approaches in the field of sequence model editing, with both introducing the concept of extensibility. But the work mechanisms of these two methods are totally different.
>
> T-patcher integrates newly neurons into model for two purposes: learn the edited data and index to determine whether the neuron is activated. The greatest distinction between SCEN and T-patcher is the two-stage training paradigm proposed by SCEN. Stage 1 focuses on learning to edit data and save weights as an expert in database, the indexing neurons in stage 2 are only used to select the corresponding experts.
>
> The existence of individual experts can ensure that new knowledge is learned with near-perfect,  while indexing neurons distinguish different experts and ensure that non-edit data is not affected. Additionally indexing neurons are not involved in the decoding process, thereby ensuring they do not disrupt the model's performance.
>
> More detailed comparative experiments can be found in Q2 and Q3 of the rebuttal for reviewer [5GUJ].
> > 2.Consider including an...
>
> To tackle this issue, we follow the findings of ROME[1] and employ FFN's $W_{down}$ as the expert. Still, for ensure the rigor of the conclusion, we also carried out additional tests, applying the llama2-7B model to the ZsRE datasets for 200 edits. The results are shown in the table below:
>
> | |Reliability|Generality|Locality|Avg|
> |--|--|--|--|--|
> |$W_{attn}$|90.0|77.3|92.0|86.4|
> |$W_{up}$|96.5|83.2|86.7| 88.8 |
> |$W_{down}$|100|90.0|83.3|91.1|
>
> According to the table, we can see that using $W_{down}$ as the expert has the best results, while $W_{attn}$ has the lowest score. This  conclusion is consistent with ROME.
>
> [1]https://arxiv.org/abs/2202.05262
> > 3.In Tables 1 and 2...
>
> Regarding the performance discrepancies observed in Tables 1&2 between Llama2-13B and 7B, we'd like to offer the following clarification:
>
> The evaluation datasets for Llama2-7B and 13B were indeed different, as we selected questions that each respective model originally failed to answer correctly. Given that Llama2-13B is a more capable model, the questions it initially failed to answer correctly tend to be more challenging. This is reflected in the evaluation data, which, for Llama2-13B, comprises more difficult questions. So this situation occurs.
>
> Additionally, we focus on compare different methods within the same model using identical data to ensure a fair comparison.

---

> > ### Author Response · Authors · 2024-06-06
> > **Request for response**
> >
> > Dear reviewer amby,
> >
> > We have posted extra experiment results (e.g., a supplemented baseline and other ablation studies) and clarifications to answer your questions. We wonder if you can let us know whether our responses address your concerns.
> >
> > Look forward to your reply.
> >
> > Best regards,
> >
> > Authors

---

### Official Review · Reviewer_67Nb · 2024-05-11

**Rating:** 7
**Confidence:** 4
**Ethics Flag:** 1

**Summary:**

This paper proposes a two-stage editing method based on expert networks, called SCEN, in the “sequential” setting that a sequence of edit requests is presented.

Overall, the proposed two-stage method, the idea of incorporating the FNN value parameters as expert networks with the indexing neurons, is interesting and novel, showing the effectiveness in the experiment settings. A work is a reasonable extension of T-patcher, towards maintaining multiple knowledge neurons and experts.
 However, the papers are not very convincing on methods and experiment settings as follows: it is not convincing why two-stage methods are required, so comparing to T-patcher, architectural advantages are not clear. Another problem is that the proposed method seems not to be scalable well, as the number of additional indexing and weight parameters is just increasing monotonically for larger scale of edits. A simple extension of T-patcher which is not in two-stage manner needs to be compared. Additionally, the paper needs to be rewritten, particularly because the method of the first stage is not very clear (how to train FNN’s down projection weight).
Nevertheless, given its positive aspects of the paper, I am currently leaning to the acceptance, while the current decision is likely changed to lower scores.

**Questions To Authors:**

Please see weaknesses. Below are further comments.
1)	Why counterfact dataset is not used, while it is used in most existing methods?
2)	The method is expanded to other multi-hop reasoning setting, like effectively working on Mquake dataset?
3)	In experiment results, only cases of 200 and 1000 edits are evaluated. How long an editing sequence where the proposed method is scalable well? Assuming that 1M or 100M edits are provided, what are the memory and inference costs in the proposed methods? In such large-scale edit, even applying Eq. (12) that selects the best expert requires non-trivial inference time.

**Reasons To Accept:**

1.	The proposed two-stage editing incorporating with the expert networks is interesting and novel, while the value parameters of FNN are considered as parameters of each expert.
2.	Experiment results show that the proposed methods are promising, comparing to other various prior methods, using the llama2 models.

**Reasons To Reject:**

1.	For each edit request, the proposed editing methods monotonically create a new indexing neuron with a weight vector, and need to keep its own FNN’s value parameters. As a result, if N number of sequential edits is provided, we need to keep N indexing neurons associated with their weight vectors as well as N FNN values parameters separately. It is largely questionable whether increasing the number of parameters is scalable in the practical situation. Even edits where their knowledge and topics are very similar will have their own separate indexing neurons and FNN value parameters, causing the method inefficient.
2.	In the two-stage method, the actual editing process is performed at the first stage, while an indexing neuron in the second stage just plays of an “indexing role” which turns on or off, depending on the input query. It is not clear why these two stages are necessary. Why the methods are not designed in a unified manner (without two-stages), like T-patcher?
3.	In the experiments, a simple extension of T-patcher not in the “two”-stage manner needs to be compared.
4.	An important part is how to actually adjust the parameters of an expert, at the first-stage. But, in Section 3.2, this part is not very clear; how to train W_down^t (down projection weight)? which layers have their own expert parameters?

---

> ### Author Rebuttal · Authors · 2024-05-29
>
> # Weaknesses:
> >1.For each edit...
>
> Our original motivation is to ensure that the edited mode work is not influenced by the non-edited data, a fact supported by our experiments. We must acknowledge that while this design aids in sequential editing and expansion, it also consumes more CPU memory space.This opens up paths for future work, like expert weight fusion. Additionally, you can find experiments about GPU usage in Q2 of the rebuttal for reviewer [4Ci8].
>
> >2.In the two-stage...
>
> The two-stage training is indeed crucial for preserving the original capabilities of the model. However, due to word limitations, you can find specific reasons and some experimental conclusions in the response to Q1 of reviewer [amby].
>
> >3.In the experiments...
>
> we have now implemented T-Patcher's method on both the llama2 7b and 13B models on the ZsRE datasets for 200 new edits. You can find detailed experimental results in the response to Q2 from reviewer [5GUJ], due to word count limitations.
>
> The results indicate that our method, SCEN, maintains a good balance between Reliability, Generality, and especially Locality, compared to T-patcher. Additionally, a comparative experiment about neuron activation incorrectly can be found in Q3 from reviewer [5GUJ].
>
> >4.An important part...
>
> In stage 1 of SCEN, the network parameters are frozen, while only the $W_{down}$ weight matrix of the FFN layer in specific layers (the 20th layer as chosen in this paper) is fine-tuned and stored as an expert. In other words, the stage 1 is the SFT that freezes most of the network parameters.
>
> # Question:
> >1.Why counterfact...
>
> >2.The method is...
>
> Our method primarily focus on sequence editing, so the comparisons are mainly with previous sequence editing approaches. We didn't specifically measure metrics against counterfact datasets for methods like ROME and MEMIT, as they aren't the sequence editing. To make comparisons more straightforward, we usually test using datasets from previous work (such as GRACE, T-patcher). Therefore, metrics from multi-hop datasets might not provide as clear comparisons.
>
>
> >3.In experiment results...
>
> Our method excels at scalable sequence editing. To illustrate, during the inference process, each edit only adds 11K to the parameters. For the Llama2-7B model, even after 10,000 continuous edits, the parameter increase is just 1.57%.
>
> Moreover, when dealing with 100M or more edits, we believe that fine-tuning the model is a more sensible approach than editing the model.

---

> > ### Author Response · Authors · 2024-06-06
> > **Request for response**
> >
> > Dear reviewer 67Nb,
> >
> > We have posted extra experiment results (e.g., a supplemented baseline and other ablation studies) and clarifications to answer your questions. We wonder if you can let us know whether our responses address your concerns.
> >
> > Look forward to your reply.
> >
> > Best regards,
> >
> > Authors

---

> > > ### Comment · Reviewer_67Nb · 2024-06-07
> > > **Reponse on authors' rebuttal**
> > >
> > > I appreciate authors' efforts to address my questions.
> > >
> > > While I now better understand the paper, the presentation part -- particularly how the parameters are modified at the first stage --- is still unclear. Also, in the case of expert network (W_down^expert & W_merge) , it is still not very convincing to me why we need to prepare two types of neurons -- neuron and weight neurons, so requiring better explaination and motivation. It seems that the use of only a single type of neuron enables to handlie the required behaviors. Otherwise, a kind of justification and abliation studies may be necessary.
> > >
> > > Overall, I like the basic idea of this paper, as a good extension of T-patcher, with promising performances.
> > > So, I have increased my recommendation score up to 7.

---

> > > > ### Author Response · Authors · 2024-06-07
> > > >
> > > > Dear Reviewer 67Nb,
> > > >
> > > > Thank you for increasing the score!
> > > >
> > > > Sorry, our reply may not be clear enough.For some unclear points, please refer to the following explanations:
> > > >
> > > > The expert network trained in the first stage is referred to as $W_{down}$. During training, we freeze the other weights and only train this particular weight. The reason for using indexing neurons is to accurately call the corresponding expert network during inference.
> > > >
> > > > The use of a two-stage process ensures non-interference and offers better performance on the dataset. In [5GUJ], it can be observed that SCEN, by adopting a two-stage process, has fewer erroneously activated neurons.
> > > >
> > > > We will further revise our paper for the camera-ready version, hoping to address all your concerns by then. We are grateful for your recommendation once again.

---

### Author Response · Authors · 2024-06-07
**Overall Comment**

We sincerely appreciate all reviewers' time and efforts in reviewing our paper. We are glad to find that reviewers generally recognize our strengths:

- The proposed two-stage editing method is **interesting and novel** [67Nb,4Ci8], the proposed expert network and index neurons for implementing selective editing methods are **promising** [amby].

- The method is **simple and a reasonable modification** of the previously proposed model editing methods [5GUJ].

- Experiments on different scales of Llama2 outperform existing model editing methods [67Nb,amby], Achieved the **best balance** among the three indicators of reliability/generalization/locality [5GUJ].

We also greatly appreciate the reviewers pointing out the shortcomings of the paper, for which we have supplemented detailed experiments and discussions. The following content is a summary of our rebuttal:

**(1). Detailed comparison with T-patcher**

- We made comparisons on Llama2-7B and 13B, and outperformed the T-patcher method on both the 7B and 13B models.

|               | Reliability | Generality | Locality |  Avg |
|---------------|:-----------:|:----------:|:--------:|:----:|
|  T-Patcher-7B |     86.0    |    76.3    |   20.5   | 60.9 |
|       SCEN-7B |     100     |    90.0    |   83.3   | 91.1 |
| T-Patcher-13B |     84.5    |    71.2    |    8.5   | 62.9 |
|      SCEN-13B |     99.5    |    80.7    |   83.5   | 87.9 |

- We also detailed the statistics of erroneous neuron activations when continuously editing 200 times with T-Patcher and SCEN. It can be seen that SCEN has fewer erroneously activated neurons compared to T-patcher.

|           | False Activation(%) | Correct Response(%) |
|-----------|---------------------|---------------------|
| T-patcher | 71                  | 48.5                |
| SCEN      | 7.5                 | 93.5                |

- We compared the GPU usage and inference time performance of T-patch and SCEN. SCEN uses less GPU memory during inference, but because it needs to schedule the expert network into GPU memory during inference, this will generate extra time consumption, so it takes longer than T-patcher.

|           | Ex-GPU(M) | In Edits(s) | Out Edits(s) |
|-----------|-----------|-------------|--------------|
| T-patcher | 121.664   | 127.7       | 126.76       |
| SCEN      | 20.996    | 303.5       | 125.32       |

**(2). Ablation Studies**

- In the paper, we use $W_{up}$ as the expert network, and we also compare the situations where other weights are used as the expert network. It can be seen that the effect is best when $W_{down}$ is used as the expert network.

|            | Reliability | Generality | Locality | Avg  |
|------------|-------------|------------|----------|------|
|  $𝑊_{attn}$ | 90.0        | 77.3       | 92.0     | 86.4 |
|  $𝑊_{up}$     | 96.5        | 83.2       | 86.7     | 88.8 |
|  $𝑊_{down}$ | 100         | 90.0       | 83.3     | 91.1 |

- For the index neurons, we demonstrated the impact of different loss composition methods on the final effect.

|                                    | Reliability | Generality | Locality | Avg  |
|------------------------------------|-------------|------------|----------|------|
| w/o $l_{disactivate}$ & $l_{margin}$ | 100         | 93.8       | 73.7     | 89.2 |
| w/o $l_{margin}$                   | 100         | 90.6       | 82.1     | 90.9 |
| SCEN loss (Eq. 8)                  | 100         | 90.0       | 83.3     | 91.1 |

**(3) Issues in the use of the dataset**

Our approach primarily focuses on sequence editing, with a specific emphasis on comparing with previous sequence editing methods. Therefore, in terms of dataset selection, we follow the datasets used in previous works.

Please contact us if we can do something else to help you better understand and recommend our paper.

---

### Decision · Program_Chairs · 2024-07-10

**Decision:**

Accept

**Comment:**

The paper introduces SCEN, a two-stage editing method for large language models that uses expert networks and indexing neurons to handle sequential edit requests. This approach enhance model performance by incorporating fine-tuned value parameters within expert networks, showing significant improvements in question-answering and hallucination reduction tasks.
Reviewers appreciate the novelty of the SCEN framework and the strong experimental results it showed. There was complaints about missing comparison to the very relavent prior work, which is addressed during the rebuttal. There are some minor concerns on limited task scope, and clarity issues in the writing.